# Prediction of malignant transformation in oral epithelial dysplasia using infrared absorbance spectra

**Barnaby G. Ellis**[1], **Conor A. Whitley**[1], **Asterios Triantafyllou**[2], **Philip J. Gunning**[3], **Caroline I. Smith**[1], **Steve D. Barrett**[1], **Peter Gardner**[4], **Richard J. Shaw**[3,5]*, **Peter Weightman**[1], **Janet M. Risk**[3]

1 Department of Physics, University of Liverpool, Liverpool, United Kingdom, 2 Department of Pathology, Liverpool Clinical Laboratories, University of Liverpool, Liverpool, United Kingdom, 3 Department of Molecular and Clinical Cancer Medicine, Institute of Systems, Molecular and Integrative Biology, University of Liverpool, Liverpool, United Kingdom, 4 Manchester Institute of Biotechnology, University of Manchester, Manchester, United Kingdom, 5 Regional Maxillofacial Unit, Liverpool University Hospitals NHS Foundation Trust, Liverpool, United Kingdom

* rjshaw@liverpool.ac.uk

**Data Availability Statement:** All files are available from the University of Liverpool Data Catalogue at https://doi.org/10.17638/datacat.liverpool.ac.uk/1622.

## Abstract

Oral epithelial dysplasia (OED) is a histopathologically-defined, potentially premalignant condition of the oral cavity. The rate of transformation to frank carcinoma is relatively low (12% within 2 years) and prediction based on histopathological grade is unreliable, leading to both over- and under-treatment. Alternative approaches include infrared (IR) spectroscopy, which is able to classify cancerous and non-cancerous tissue in a number of cancers, including oral. The aim of this study was to explore the capability of FTIR (Fourier-transform IR) microscopy and machine learning as a means of predicting malignant transformation of OED. Supervised, retrospective analysis of longitudinally-collected OED biopsy samples from 17 patients with high risk OED lesions: 10 lesions transformed and 7 did not over a follow-up period of more than 3 years. FTIR spectra were collected from routine, unstained histopathological sections and machine learning used to predict malignant transformation, irrespective of OED classification. PCA-LDA (principal component analysis followed by linear discriminant analysis) provided evidence that the subsequent transforming status of these 17 lesions could be predicted from FTIR data with a sensitivity of 79 ± 5% and a specificity of 76 ± 5%. Six key wavenumbers were identified as most important in this classification. Although this pilot study used a small cohort, the strict inclusion criteria and classification based on known outcome, rather than OED grade, make this a novel study in the field of FTIR in oral cancer and support the clinical potential of this technology in the surveillance of OED.

## Introduction

Oral squamous cell carcinoma (OSCC) has a worldwide incidence rate of over 370 000 [1] and a 5-year survival rate that remains less than 60% [2]. It is often preceded by a spectrum of

**Funding:** JMR, RJS, PW & SDB: Cancer Research UK grant number C7738/A26196. https://www.cancerresearchuk.org/ BGE & CAW were supported by Engineering and Physical Sciences Research Council (EPSRC) PhD studentships. https://epsrc.ukri.org/ The funders had no role in the study design, data collection and analysis, decision to publish, or preparation of the manuscript.

**Competing interests:** The authors have declared that no competing interests exist.

clinical changes, collectively termed potentially pre-malignant oral epithelial lesions or oral potentially malignant disorders (OPMDs) [3]. OPMDs include white patches (leukoplakia), red patches (erythroplakia) and mixed color patches (erythroleukoplakia) of the oral mucosa and are usually investigated by biopsy and histopathological examination. The latter typically shows morphological changes of the surface squamous epithelium, which are conveniently described as oral epithelial dysplasia (OED), that include variable cellular atypia, proliferative activity and loss of normal patterns of differentiation. The World Health Organization (WHO) defines OED as "altered epithelium with an increased likelihood for progression to squamous cell carcinoma" [4]. The malignant potential (progression to invasive tumor) of OPMDs range from as low as 0.13% in some leukoplakias [5] to >50% in some erythroplakias [6]; a meta-analysis of OED data indicates a malignant transformation rate of 12% within 2 years, increasing to 22% within 5 years [7].

The histopathological grading of OED as mild, moderate or severe is widely popular and time-honored, based on various architectural and cytological changes, and is endorsed and formalized by WHO [8]. Other grading systems, including a binary "high/low risk" scheme [9], have been proposed, but the standard across the field remains the three-tiered scheme. Numerous studies [10–12] have shown a significant relationship between the histopathological grade and risk of malignant transformation, however there are a similar number of conflicting reports which suggest a much less direct relationship, highlighting other risk factors [13–15]. Although biopsy and histopathological assessment of OPMDs forms the basis of clinical management, the grading of OED is influenced by inter- and intra-observer variations [9], reflecting the subjectivity of the process, and improvements are required.

Given its clinical significance, OED has been investigated by a wide spectrum of predominantly biology-based methodologies, but none of the proposed biomarkers for predicting risk are in routine clinical use [16]. Less attention has been paid to the application of alternative methodologies utilizing the chemical or physical properties of the cells. Among the alternative methodologies, those based on vibrational spectroscopy have been increasingly introduced to biomedical research [17, 18]. Fourier transform infrared (FTIR) spectroscopy utilizes infrared (IR) light over a broad spectral range to assess the overall chemical profile of a sample. Molecules which vibrate at frequencies corresponding to the wavelengths applied will absorb the radiation at those wavelengths, resulting in an absorption spectrum characteristic of the chemical moieties present. FTIR micro-spectroscopy (FTIR-MS) combines IR spectroscopy with precise spatial information enabling the rapid acquisition of hyperspectral images directly related to the location and distribution of chemical components, for example in tissue samples. Hyperspectral data acquired using FTIR-MS is highly dimensional as each raw spectrum, obtained from a region approximately 5 μm x 5 μm in size, contains at least $10^3$ absorption variables. Subtle differences between tissue areas are concealed by dominant common features in chemical composition, necessitating the use of sophisticated numerical approaches to extract useful information [19]. Common modelling methods whose aim is to reduce the complexity of the dataset include principal component analysis (PCA) [20] and linear discriminant analysis (LDA) [21, 22].

FTIR-MS has been utilized in biomedical research, with a particular focus on its application to the investigation of cancerous tissues (reviewed in ref. [23]). Our own recent data suggests that this methodology is applicable to OSCC [24], and other studies have successfully associated vibrational spectroscopy data with the contemporaneous histopathological classification of potentially malignant oral lesions [25, 26]. The present investigation takes a different approach, using a supervised, retrospective analysis of tissue samples from high risk OED lesions from patients with prolonged, longitudinal clinical follow-up and known outcome (transformation or no transformation) to explore the capability of FTIR-MS and machine learning as a means of predicting malignant transformation of OED.

## Methods

Seventeen patients with biopsy-proven OED were included in this study (Table 1). All patients were part of a larger cohort for whom the clinical determinants of transformation have been described [15] and had given written informed consent to a UK NHS Research Ethics Committee approved study that was run in compliance with the Helsinki Declaration (Liverpool Central REC ref: EC 47.01). Patient selection was limited by inclusion of only lesions with a histopathological diagnosis of moderate or severe grade OED, absence of previous OSCC, at least 42 months follow-up from the time of biopsy, and the availability of relevant archival formalin-fixed paraffin-embedded (FFPE) tissue. Although inclusion was partly dependent on a number of non-clinical factors such as availability of samples, the group of patients used in this present study remained representative of the total cohort [15]. Thus, there was a higher female:male ratio in the transforming group, which also had a preponderance of lateral tongue lesions, and there were proportionally more smokers in the non-transforming group (Table 1). It should be noted that there was no significant difference between the distribution of severe and moderate grade OED lesions in the two groups.

A single archival FFPE tissue block containing incisional biopsy material was obtained from each of 10 patients at the closest timepoint to transformation (range 2–43 months prior to transformation) (T lesions; Table 1). A single archival FFPE block containing incisional biopsy material with more than 43 months transformation-free follow-up from the date of biopsy (range 43–108 months) was obtained from each of 7 patients (NT lesions; Table 1). None of these NT lesions had been excised during the follow-up period.

From each of the 17 FFPE blocks, four adjacent 5 μm tissue sections were obtained, reserving the first and last for routine deparaffinization, hematoxylin and eosin (H&E) staining and histopathological re-examination. The intervening two paraffinized, unstained sections were mounted on separate 20 mm diameter calcium fluoride (CaF$_2$) disks for IR imaging experiments. An area of dysplasia corresponding to the histopathologically most extreme OED present was identified in each H&E-stained section and marked as the target for FTIR spectroscopy. This area was termed the region of interest (ROI).

FTIR imaging data were acquired using a Varian 620 microscope coupled to an Agilent Cary 670 spectrometer (Agilent, Stockport, UK) enclosed within a purging chamber to eliminate water vapor and carbon dioxide contributions. The instrument was configured to collect mid-IR transmission data between 900 and 3800 cm$^{-1}$ with a spectral resolution of 4 cm$^{-1}$ and pixel size of 5.5 μm, allowing the simultaneous acquisition of 128 x 128 spectra over a field of view of approximately 0.5 mm$^2$. CaF$_2$ disks with mounted sections were loaded onto a 3D-printed slide holder capable of containing three disks. Disks were imaged two at a time, with the third position reserved for a clean, blank disk to allow for microscope calibration and spectral background correction. The semiconductor detector (Mercury-Cadmium-Telluride) in the FTIR microscope was cooled with liquid nitrogen to 78 K in order to reduce thermal noise in the data. Using the average of 128 scans of the blank disk for background correction, a hyperspectral IR image was obtained for each ROI by averaging 64 scans of the identified dysplastic area. A built-in mosaic function was utilized for cases where the extent of the surface oral epithelium in the tissue sections could not be visualized in one field of view of the microscope and enabled the acquisition of larger, composite images.

Subsequently, the FTIR images were cross-referenced with scanned images of the corresponding H&E sections to confirm the location and extent of dysplasia within the ROI (Fig 1A). To annotate IR spectral data originating in dysplastic epithelium, each hyperspectral image was subjected to a two-tiered, k-means cluster analysis (Fig 1A). Initially, the epithelium was identified by a specialist, head and neck histopathologist (AT). The first clustering step

**Table 1. Patient and sample cohort characteristics.**

| Patient group | Identifier | Age at biopsy | Gender | Site | Number of clinical sites | Clinical Presentation | Lesion size (mm$^2$) at presentation | Histology Grade of ROI[a] | Time before transformation (months) | Time cancer free (months) | Lifestyle tobacco | Lifestyle alcohol |
|---|---|---|---|---|---|---|---|---|---|---|---|---|
| **Transforming** n = 10 | 12089 | 58 | M | Ventral Tongue | single | Erythroleukoplakia | 101–500 | severe | 2 | | y (2-20py)[b] | y |
| | 12201 | 52 | M | Buccal | single | Leukoplakia | >500 | severe | 4 | | y (>20py) | y |
| | 12260 | 74 | F | Floor of Mouth | single | Leukoplakia | ≤100 | moderate-severe | 5 | | y (>20py) | y |
| | 12127 | 85 | M | Lateral Tongue | single | Leukoplakia | 100–500 | moderate | 7 | | n | n |
| | 12257 | 49 | F | Lateral Tongue | single | Leukoplakia | 100–500 | moderate | 12 | | n | y |
| | 12248 | 69 | F | Lateral Tongue | multiple | Erythroleukoplakia | 100–500 | severe | 14 | | n | n |
| | 12263 | 45 | F | Lateral Tongue | single | Leukoplakia | 100–500 | moderate-severe | 18 | | y (5-20py) | y |
| | 12104 | 70 | F | Lateral Tongue | single | Erythroleukoplakia | 100–500 | moderate | 26 | | n | n |
| | 12195 | 45 | F | Ventral Tongue | multiple | Erythroleukoplakia | >500 | moderate | 33 | | y (5-20py) | y |
| | 12219 | 68 | F | Ventral Tongue | single | Leukoplakia | ≤100 | moderate-severe | 43 | | y (5-10py) | n |
| **Non-Transforming** n = 7 | 12181 | 49 | F | Soft Palate | single | Leukoplakia | >500 | severe | | 158 | n | y |
| | 12330 | 47 | F | Soft Palate | multiple | Erythroleukoplakia | >500 | moderate | | 108 | y (5-20py) | y |
| | 12098 | 78 | M | Ventral Tongue | single | Leukoplakia | >500 | moderate | | 106 | y (>20py) | y |
| | 12332 | 71 | F | Mandibular alveolus | multiple | Leukoplakia | ≤100 | moderate | | 91 | n | y |
| | 12329 | 59 | M | Floor of Mouth | single | Erythroleukoplakia | ≤100 | moderate | | 75 | y (>20py) | y |
| | 12162 | 61 | F | Floor of Mouth | single | Erythroleukoplakia | ≤100 | severe | | 67 | y (>20py) | y |
| | 12141 | 47 | F | Floor of Mouth | multiple | Erythroleukoplakia | ≤100 | severe | | 43 | y (>20py) | y |

[a]ROI = region of interest: the target for FTIR spectroscopy

[b]py = pack years (= packs per day multiplied by years smoked).

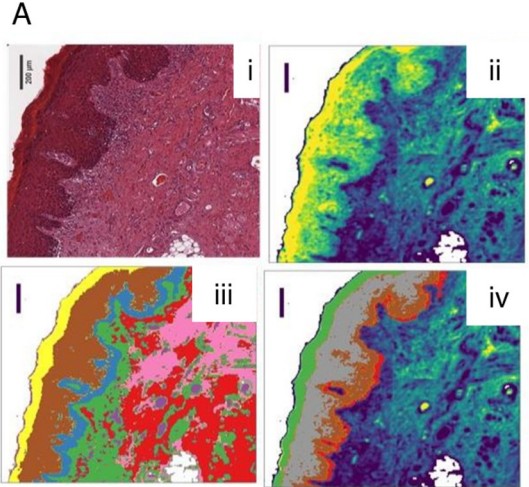

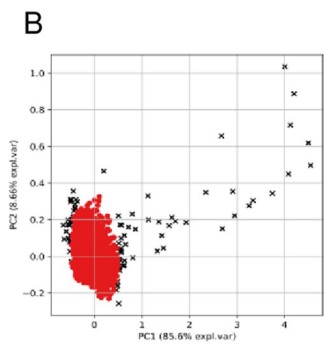

**Fig 1. Identification of IR data to be used in classification.** (A) Example of two-tiered k-means cluster analysis. (i): H&E image; (ii): corresponding FTIR hyperspectral image; (iii): the first tier of k-means cluster analysis identifies the surface epithelium as 3 separate, spectrally similar regions (identified as yellow, brown and blue colored layers by histopathological comparison with (i)). Histologically, the blue colored cluster broadly corresponds to the basal layer; the brown cluster to parabasal and prickle-cell (spinous) layers; and the yellow to the keratinized layers; (iv) the second tier of k-means cluster analysis subdivides the epithelium into four clusters of spectrally similar regions (green, grey, brown & orange). This second 2-tier clustering appears to separately identify the parabasal (brown) and spinous (grey) layers. Histopathology plus PCA clustering of FTIR data selects the brown and red clusters for use in modelling. Scale bar = 200 μm. (B) Illustration of the quality control process. Spectra identified as lying outside the 95% confidence interval by the Hotelling's T-squared test (black crosses) were removed from dataset. Data in this figure were obtained from the same tissue section as in part (A).

was then used to identify this structure based on its IR hyperspectral profile. The data from this first clustering step was then processed during the second clustering in order to identify regions within the epithelium based on their IR hyperspectra. Histologically, dysplasia was often centered on the basal and parabasal layers and, in the case of more severe dysplasia, in the upper prickle-cell layer. IR data from areas with both a histological assessment of dysplasia and where k-means clustering identified relative chemical homogeneity were selected for modelling.

Spectra that originated from dysplastic material in each IR hyperspectral image were then subject to an initial quality check to discard anomalous spectra. This involved using PCA as a tool to decompose the spectra from each image into five principal components, and then employing Hotelling's $T^2$ summary statistic to determine which spectra lie furthest away from the origin, discarding any which lie outside the 95% confidence interval [27]. The remaining spectra were retained for modelling. Although the FTIR and H&E images had been cross-referenced in order to locate the imaged dysplastic epithelial layers, the spectral data were grouped into two categories based on the clinical outcome of the lesion from which they were taken regardless of OED severity: T lesions underwent malignant transformation and NT lesions did not undergo transformation.

We have developed an objective optimization framework, PipeOpt (paper in preparation), that aims to maximize the efficiency of a classifier by optimizing the parameters of each pre-processing step of the IR data, determining their ideal sequence and identifying the best performing classification method(s) using Bayesian optimization (Table 2). This process probabilistically converges on the best hyperparameters for each unique series of pre-processing and classification steps (defined as a pipeline) by iteratively updating the associated Matthew's correlation coefficient [28, 29]: a statistic scaled between -1 and 1 which includes consideration of

**Table 2. Pre-processing steps tested in the PipeOpt objective optimization framework.**

| Step | Method | Hyperparameter | Options |
|------|--------|----------------|---------|
| Smoothing | Savitzy-Golay (SG) smoothing | Window size | 5,7,9,11,13,15,17,19,21 |
| | PCA | Explained Variance | 80–95% |
| | none | | |
| Baseline Correction | Rubberband | N/A | Y/N |
| | SG Differentiation | Window size (if no smoothing) | 5,7,9,11,13,15,17,19,21 |
| | | Polynomial order | 2,3 |
| | | Differentiation order | 1,2 |
| | none | | |
| Paraffin Correction | Removal of the spectral region dominated by paraffin wax (1340–1490 cm$^{-1}$) | | |
| Normalisation | Vector | N/A | Y/N |
| | Min-max | N/A | Y/N |
| | Amide I | N/A | Y/N |
| | none | | |
| Scaling | Standard scaling | N/A | Y/N |
| | Min-max | N/A | Y/N |
| | none | | |
| Feature Extraction | PCA | Explained Variance | 90–98% |
| | none | | |
| Classifier | Logistic regression | Regularisation strength | 0.001–10 |
| | Linear discriminant analysis | N/A | Y/N |
| | Random Forest | Max-depth | Y/N |
| | | Minimum samples per split | 2,3,4,5 |
| | | Minimum samples per leaf | 1,2 |
| | | Bootstrap | Y/N |

Step: typical pre-processing step utilized in analyzing IR data

Method: methods typically used to perform each pre-processing step

Hyperparameter: typical parameters associated with each method

Options: typical options for each parameter. Each step (apart from Paraffin correction and Classification) also has a bypass option and the steps may be performed in any order (except for classification).

Bayesian optimization was used to identify hyperparameter options for the resulting 3 x 3 x 1 x 4 x 3 x 2 x 3 = 648 pipelines.

both the sensitivity and specificity of the resulting classification. This allows for faster convergence to the ideal compared to an approach that samples every possible combination of hyperparameters. To assess the performance of each pipeline, 70 training sets were created by using a leave-one-pair-out cross validation (LOPOCV) method: pairs of samples (comprising 1 T and 1 NT sample in every possible combination) was set aside as the test set, while data from the remaining 6 NT and 9 T samples was used for training. Equal numbers of spectra (n = 500) were used from each sample to avoid sample-related bias that might influence the optimization and the optimal pipeline was defined as that with the highest mean Matthew's correlation coefficient.

Following determination of the optimal pipeline, the same sequence of pre-processing and classifier steps were used to analyze all of the available data (range 3891–5437 spectra) from the 17 samples using the same LOPOCV routine as before to create training and test sets.

PCA-LDA is a feature extraction and classifier hybrid that uses a series of linear transformations to decompose the data from absorption variables to a single variable called a linear

discriminant (LD), the value of which is called the LD score. The LD score is dependent upon the PCA and LDA loading vectors, which are a measure of the relative weight that each wavenumber in the spectrum contributes to that part of the analysis. The LD scores for each datapoint in every lesion were plotted against their frequency of occurrence in relation to their known transformative capacity and were also used to identify key wavenumbers in the discrimination of T from NT datapoints.

At each iteration of train/test, the predicted outcome (transformation or no transformation) at every datapoint for the test sample was determined and the lesion as a whole classified as T or NT based on a simple majority.

The FTIR data was converted from native data format using the ChiToolbox package MATLAB [30]. All data transformation and statistical analyses were performed using either in-house developed packages (PipeOpt) or third-party packages implemented in Python v3.9.

## Results

Each of 648 different data pipelines were tested 70 times, each iteration having a different pair of samples (one T and one NT) removed to create the test/train datasets, and the mean Matthew's correlation coefficient across these 70 iterations was determined (Fig 2). The pipeline

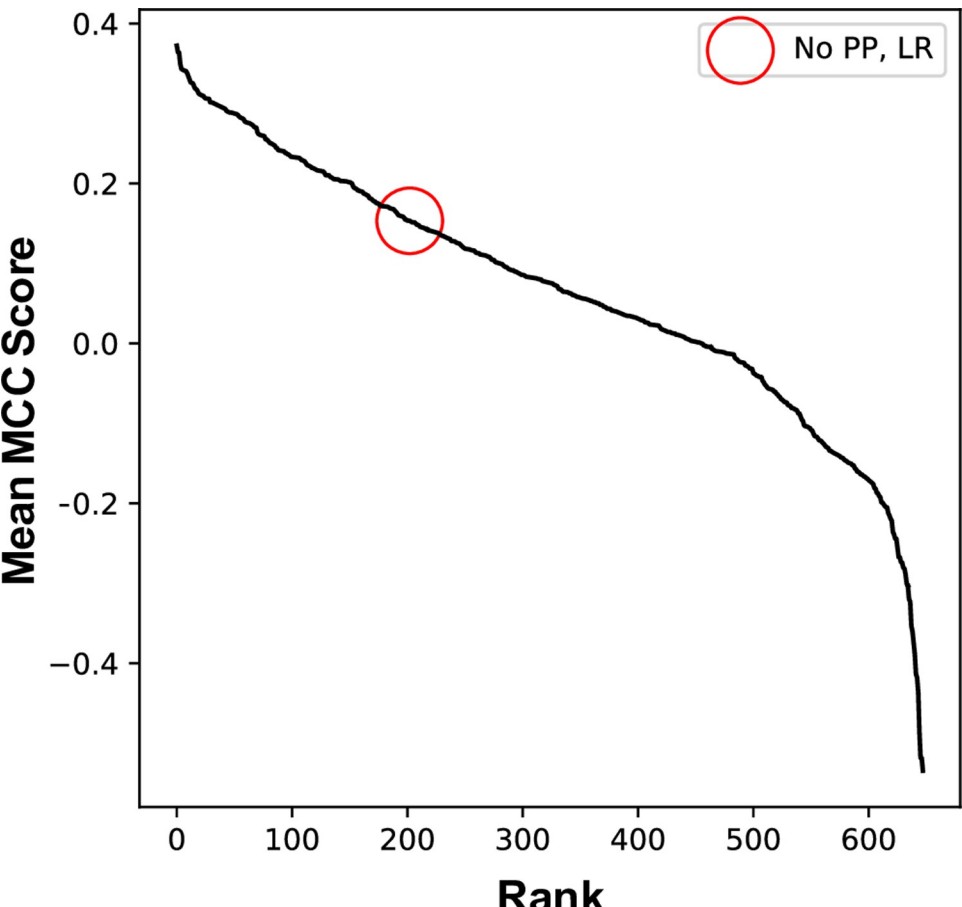

**Fig 2. Determination of the optimal analysis pipeline for this dataset.** Mean of Matthew's correlation coefficient (MCC) for each of 648 analysis pipelines generated from the dysplasia dataset plotted in descending order. Circle identifies the MCC for the pipeline with no data pre-processing and classification using linear regression (MCC = 0.15).

with the highest mean correlation coefficient (0.37) was identified from this analysis and is as follows: denoising of the spectra using the Savitzky-Golay smoothing algorithm [31] with a window size of 15 and polynomial order of 2; first order differentiation of the spectra to remove effects such as scattering and background interferences; removal of the spectral region dominated by paraffin wax (1340–1490 cm$^{-1}$) [23]; normalization so that the sum of the squares of each spectrum is equal to 1 (vector normalization) to account for variations in sample thickness; PCA-LDA classification. In this analysis, PCA was applied to the spectral data to decompose it into the number of principal components that described 90% of the explained variance in the original dataset and LDA was then used to discriminate between the two groups of lesions (T and NT) using the principal components as input.

The mean sensitivity of this discriminatory model when applied to the whole dataset at the level of an individual spectrum (i.e. each datapoint in every sample taken as an individual element) was 74 ± 2.8% and the specificity was 69 ± 3.2%, while the mean and median receiver operator characteristic (ROC) further demonstrated the performance of the model (Fig 3A). Moreover, the PCA-LDA-derived linear discriminant score showed good separation of the two classes (T and NT) (Fig 3B) and the weighting assigned to different wavenumbers during this analysis, allowed the provisional identification of six wavenumbers that provided the most discriminatory power (Fig 3C).

However, for clinical utility, the transformative capacity of the whole lesion is more relevant than that for each individual datapoint. Therefore, in each iteration of train/test, the predicted outcome (transformation or no transformation) at every datapoint for the test sample was used to define the lesion as T or NT based on a simple majority (i.e. ≥50% of datapoints). The sensitivity per lesion was 79 ± 4.9% and the specificity was 76 ± 5.1%. However, the prognosis of some lesions was better predicted than others (Fig 4), with 2 T lesions and 1 NT lesion being incorrectly predicted in ≥50% of the test/train iterations in which they were the test sample.

To better visualize this observation, the probability of transformation for each datapoint from 4 lesions was color coded and mapped back onto a representation of the whole section (Fig 5). Lesions that were most accurately predicted showed homogeneous areas of correctly labelled datapoints with only a few incorrectly labelled points (Fig 5B and 5F). Conversely, lesions that were less accurately predicted demonstrated some areas that were predicted to transform and some that were predicted to not transform (Fig 5D and 5H). The OED grade of the lesions did not appear to correlate with the success or otherwise of the prediction of transformation.

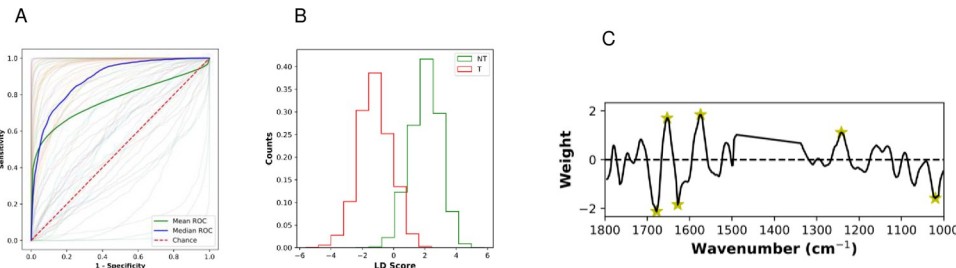

**Fig 3. Performance of the model taking each datapoint individually.** Means from 70 iterations of train/test sets are presented. (A) Mean (green) and median (blue) receiver operator characteristic (ROC) curves. Red dotted line would be achieved by random chance. Pale lines are individual ROC curves for each iteration; (B) Histogram of frequency of occurrence of linear discriminant (LD) scores (counts) plotted for all datapoints from transforming (T: red) and non-transforming (NT: green) lesions, showing separation of the two classes; (C) Plot showing the weighting (a measure of relative importance) assigned to each wavenumber during the PCA-LDA analysis. Features marked with a yellow star show the largest magnitude in weighting: 1678 cm$^{-1}$, 1653 cm$^{-1}$, 1628 cm$^{-1}$, 1574 cm$^{-1}$, 1242 cm$^{-1}$ and 1020 cm$^{-1}$.

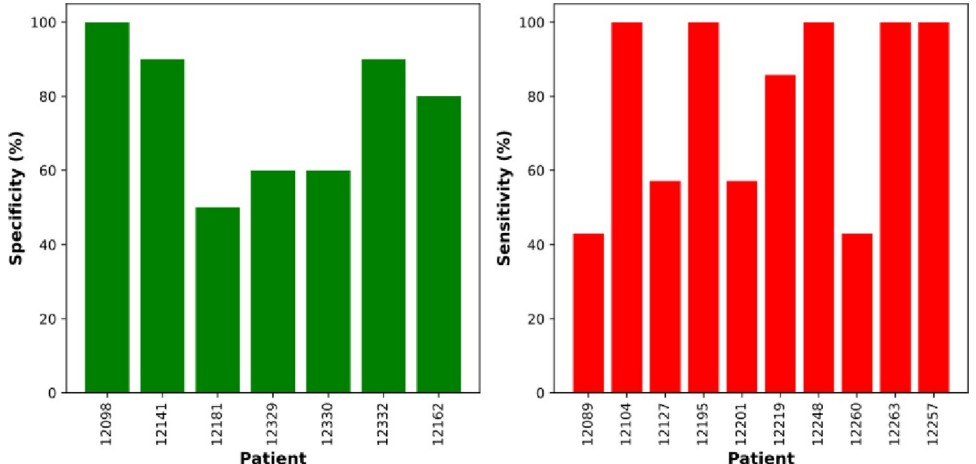

**Fig 4. Performance of the model at the lesion level.** Means from 70 iterations of train/test sets are presented. Frequency at which each whole lesion was correctly predicted when it appeared in the test set (based on LOPOCV, each NT lesion is present in 10 test pairs and each T lesion in 7 test pairs). Data is shown for each individual NT (left) and T (right) lesion and plotted as specificity (true negatives) and sensitivity (true positives). Patient numbers are research sample IDs.

## Discussion

We have applied machine learning to infrared data collected by FTIR-MS from a number of high risk OED lesions with known transforming potential. This is in contrast to most other studies analyzing FTIR data that has been collected from oral premalignant lesions, which commonly correlate IR data with OED stage rather than outcome. The process correctly predicted the capacity to transform with a sensitivity of 79 ± 4.9% and a specificity of 76 ± 5.1%, which is better than when OED grade alone is used. A histological grade of severe OED or carcinoma-in-situ has been observed to have a significantly increased malignant transformation rate compared with mild or moderate OED (P<0.008) [7], but this grading is still only

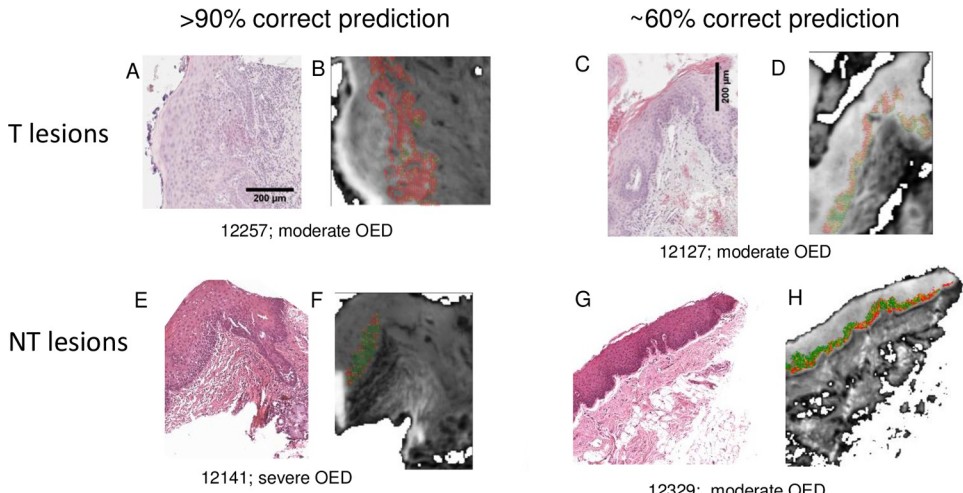

**Fig 5. Representative images demonstrating identification of lesional areas that are mis-labelled.** Every datapoint for each lesion was color coded to represent the probability of transformation (T: red; NT: green; see color bar) and mapped back onto a representation of the whole section for two T (top) and two NT (bottom) lesions. Lesions predicted with high (left) and lower (right) accuracy are shown. (A), (C), (E) and (G): H&E images; (B), (D), (F) and (H): corresponding maps of predicted datapoints on IR images.

predictive for 24–40% of such lesions [7, 32]. It is of note that, in the limited cohort used in the present study, equivalent numbers of severe and moderate OED were present in the T and NT groups, and OED grade did not correlate with the ability of the classifier to accurately predict transformation potential.

Two lesions known to transform were predicted to be non-transforming in the current analysis. Based on our knowledge of oral cancer development, it may be hypothesized that the lesions are most probably heterogeneous with areas possessing transforming potential and areas without this potential. Thus, the biopsy may not have been representative of the region that underwent subsequent transformation. Similarly, given the relatively large size of the region used for IR imaging compared to the size of individual cells, the IR imaged area will contain areas displaying an 'IR transformation fingerprint' and areas that do not. In the analysis presented here, it is the predominant fingerprint (i.e. $\geq$50% of datapoints) that was used to classify the lesion as a whole, but, clinically, the worst area rather than the predominant signal may be more important when predicting transformation and selecting appropriate clinical treatment. Future development of this method would investigate how altering the 50% threshold affects the capability of the model, as the sensitivity would be expected to increase as the threshold is reduced but at the expense of specificity and vice versa. This is an important area for clarification in a larger study and is related to clinical needs. Increased sensitivity (i.e. better prediction of transforming status) could lead to decreased surveillance intervals, treatment such as excision or enrolment onto chemoprevention trials. Conversely increased specificity (i.e. better prediction of non-transforming status) might lead to changes in clinical practice to allow for safe discharge or increased follow-up intervals. The identification of areas predicted to have transforming capacity in one NT lesion is more difficult to explain, as none of the NT lesions were excised during the follow-up period. However, it might be hypothesized that very small islands of putatively transforming OED could have been completely excised during the biopsy procedure, unintentionally performing a therapeutic excisional biopsy despite the intention for diagnostic incisional biopsy.

Pre-processing of FTIR data is acknowledged to improve the performance of subsequent classification models [33], but the choice of both the protocol and the subsequent modelling method is often highly subjective and its efficacy is dependent on the characteristics of the dataset. Instead of applying pre-processing steps in an arbitrary manner, a novel objective optimization method was used in the current study to maximize the efficiency of the OED transformation classifier by optimizing the parameters of each pre-processing step of the IR data, determining their ideal sequence and identifying the best performing classification method(s), or pipeline, using Bayesian optimization. Considering that the trialed pipelines all contain theoretically sensible pre-processing and classifier combinations, the significant variation in Matthew's correlation coefficient score (-0.53 to 0.37) was surprising. It was noted, however, that many of the negative correlations were obtained when data was not normalized. Infrared data are very sensitive to the amount of substance being probed, so mitigating for inevitable variability in sample thickness and preparation is crucial in order to build models derived from multiple specimens.

The decision to use a LOPOCV strategy was taken because the small sample size precluded the division of specimens into larger test/train sets. LOPOCV produces the most robust estimation of the model's true performance, since every combination of patients is used in the training and testing of the model. However, the small sample size (n = 17) led to a high standard deviation in the mean sensitivity, specificity and ROC variance at the individual datapoint level because biological variation both between patients and within individual lesions is to be expected and will impinge on the analysis under a LOPOCV strategy. Thus, a larger, multi-center study is required to test the model further.

Four of the six wavenumbers attributed the most weight during this classification can be assigned to components of the amide I and II bands [34, 35], which are situated at 1700–1600 cm$^{-1}$ and 1600–1500 cm$^{-1}$ respectively. Absorbance at these wavenumbers can be directly attributed to the vibrating modes of repeating peptide bonds, but the convoluted nature of the amide bands renders the task of attributing specific moieties to particular wavenumbers difficult. The remaining features with the highest weight, and hence discriminating power, are at 1242 cm$^{-1}$ and 1020 cm$^{-1}$: regions known to be dominated by contributions from DNA, RNA and glycogen [34]. Thus, a relatively high weighting of the IR data obtained at 1242 cm$^{-1}$ might be indicative of an increase in DNA aneuploidy in T lesions compared with NT lesions, a known early event in oral carcinogenesis [36, 37]. Similarly, the characteristic absorption peak of glycogen is centered at approximately 1030 cm$^{-1}$ and its importance in the experiments presented in this report may correlate with the observation that abundance of the molecule is depleted in pre-malignant tissue as a result of increased proliferation requiring additional energy [38]. This association with glycogen depletion has been applied in the use of Lugol's Iodine staining in an attempt to identify and clear OED at the margins of oral cancer resections [39]. Despite these reassuring correlations between wavenumbers with high weighting in the discriminatory model and previously recognized biological observations, it should be remembered that the absorbance of IR light at any particular wavenumber by a biological tissue is the sum of the absorbance by a number of different biochemical molecules, and it should not be expected that a multivariate analysis combining DNA aneuploidy and glycogen levels will be as effective a discriminator as the model presented here. Future research with larger sample numbers and intention-to-treat biopsy specimens should use multivariate analysis to assess how many key wavenumbers are necessary, in conjunction with clinicopathological variables, to build a clinically useful discriminatory model. This reduction in the number of wavenumbers required for discrimination will, in turn, lead to the development of less expensive IR-based technology, perhaps utilizing quantum cascade lasers (QCLs) [40, 41], that might be employed in routine pathology laboratories.

## Conclusions

This study of a pathologically defined set of OED specimens with known outcome suggests that the analysis of IR data can distinguish lesions with the capacity to transform to oral cancer from those that do not, regardless of OED grade. This represents a novel analysis of FTIR data collected from oral premalignant lesions, as data is commonly correlated with OED stage rather than outcome. The results are encouraging, bearing in mind the small sample size and the inherent clinical and biological limitations of using a small biopsy to reflect a much greater field of potential malignant change, and may come to represent a step forward in the clinical assessment of such lesions that allows improved treatment planning. Further research should concentrate on increasing sample size and complexity to reflect the clinical conundrum and the development of technology to apply the methodology in a timely and cost-effective manner.

## Author Contributions

**Conceptualization:** Richard J. Shaw, Peter Weightman, Janet M. Risk.

**Data curation:** Barnaby G. Ellis.

**Formal analysis:** Barnaby G. Ellis, Conor A. Whitley, Steve D. Barrett.

**Funding acquisition:** Steve D. Barrett, Richard J. Shaw, Peter Weightman, Janet M. Risk.

**Investigation:** Barnaby G. Ellis, Asterios Triantafyllou, Philip J. Gunning, Caroline I. Smith, Peter Gardner.

**Methodology:** Barnaby G. Ellis, Janet M. Risk.

**Project administration:** Caroline I. Smith, Janet M. Risk.

**Resources:** Barnaby G. Ellis, Asterios Triantafyllou, Philip J. Gunning, Peter Gardner, Janet M. Risk.

**Software:** Barnaby G. Ellis, Conor A. Whitley.

**Supervision:** Steve D. Barrett, Richard J. Shaw, Peter Weightman, Janet M. Risk.

**Visualization:** Barnaby G. Ellis.

**Writing – original draft:** Barnaby G. Ellis, Janet M. Risk.

**Writing – review & editing:** Barnaby G. Ellis, Conor A. Whitley, Asterios Triantafyllou, Philip J. Gunning, Caroline I. Smith, Steve D. Barrett, Peter Gardner, Richard J. Shaw, Peter Weightman, Janet M. Risk.

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
