## [Decision Letter · Decision Letter 0]

31 Jan 2022

PONE-D-21-39833Prediction of malignant transformation in oral epithelial dysplasia using infrared absorbance spectraPLOS ONE

Dear Dr. Shaw,

Thank you for submitting your manuscript to PLOS ONE. After careful consideration, we feel that it has merit but does not fully meet PLOS ONE’s publication criteria as it currently stands. Therefore, we invite you to submit a revised version of the manuscript that addresses the points raised during the review process.

Please respond to the reviewers' comments. It sounds like an intriguing study but would gain from some clarification.

We look forward to receiving your revised manuscript.

Kind regards,

Patrick Ha

Academic Editor

PLOS ONE

Journal Requirements:

[This study was funded by Cancer Research UK C7738/A26196. BGE and CAW were supported by Engineering and Physical Sciences Research Council (EPSRC) PhD studentships.]

 [JMR, RJS, PW & SDB: Cancer Research UK grant number C7738/A26196. https://www.cancerresearchuk.org/ BGE & CAW were supported by Engineering and Physical Sciences Research Council (EPSRC) PhD studentships. https://epsrc.ukri.org/

The funders had no role in the study design, data collection and analysis, decision to publish, or preparation of the manuscript.]

Reviewers' comments:

Reviewer's Responses to Questions

**Comments to the Author**

1. Is the manuscript technically sound, and do the data support the conclusions?

Reviewer #1: Yes

Reviewer #2: Partly

2. Has the statistical analysis been performed appropriately and rigorously? 

Reviewer #1: Yes

Reviewer #2: Yes

3. Have the authors made all data underlying the findings in their manuscript fully available?

Reviewer #1: Yes

Reviewer #2: Yes

4. Is the manuscript presented in an intelligible fashion and written in standard English?

Reviewer #1: Yes

Reviewer #2: Yes

5. Review Comments to the Author

Reviewer #1: This article describes a retrospective review of the use of Fourier-transform infrared (FTIR) absorbance spectra in predicting the transformation of oral epithelial dysplasia into oral carcinoma.

The authors compare spectroscopic data from 10 patients with oral epithelial dysplasia that underwent transformation into carcinoma to that from 7 patients with oral epithelial dysplasia that did not undergo transformation. The authors then use PCA-LDA (principal component analysis followed by linear discriminant analysis) to demonstrate that the FTIR data predicted malignant transformation with a sensitivity of 79% and specificity of 76%.

While I can’t comment on the validity of the methodology in this study due to my lack of expertise in IR spectroscopy, my impression is that this manuscript thoroughly describes a potentially compelling method of investigating oral epithelial dysplasia. This is of particular clinical relevance given that degree of dysplasia poorly predicts transformation. I therefore recommend acceptance.

Reviewer #2: In this study by Ellis et. al., the authors retrospectively review 17 patients with high risk oral epithelial dysplasia, 10 of which transformed to malignancy (T=transformed) and 7 of which did not (NT=non transformed), in order to determine whether Fourier-transformed IR microscopy and machine learning algorithm can predict malignant transformation.

The authors utilize PCA and k-means cluster analysis, with outliers removed by Hotellings T^2 summary statistic, to determine the spectral data to include in the development of the optimal machine learning algorithm. Having identified the data to include, the authors then use an unpublished optimization framework, PipeOpt, to test the performance of each pipeline using all possible combinations of pairs of T/NT samples as the test set, with the remaining samples as the training set. In total, 648 unique pipelines were tested. The PCA-LDA-derived linear discriminant score and weighting for different wavenumbers ultimately identified the 6 wavenumbers with the highest discriminatory power between T and NT groups. When applying this machine learning algorithm to infrared data collected by FTIR-microscopy, the sensitivity was 79%, specificity 76%, in identifying which lesions would undergo malignant transformation.

Overall, this study aims to address an important and difficult clinical scenario of which pre-malignant oral cavity lesions ultimately undergo malignant transformation with novel methodology. I am unable to comment on the mathematical soundness of the determination and implementation of the pipeline analysis. However, there are several areas of major concern, that if addressed, could eventually make this work suitable for publication.

1. From a clinical perspective, while it is helpful to understand which OED with severe histopathology undergo malignant transformation for counseling purposes, from an intervention standpoint, this is less relevant as most surgeons will excise forms of severe dysplasia. It would be of greater clinical utility to also include lesions that are mild and/or moderate dysplasia as a separate analysis, as there is more ambiguity in clinical decision making.

2. The authors do acknowledge that the small sample size is a limitation of this study. However, with testing of 648 different pipelines and 17 samples, there is a concern that the authors are overfitting their data, thereby making it difficult to interpret the utility of this algorithm. It would be helpful to see the predictive accuracy of this methodology replicated with a validation cohort.

3. Did all lesions undergo only incisional biopsy and then were followed? Or did some of these lesions undergo curative surgical excision?

4. Please clarify how the region of interest was determined.

6. PLOS authors have the option to publish the peer review history of their article (what does this mean?). If published, this will include your full peer review and any attached files.

Reviewer #1: No

Reviewer #2: No

---

## [Author Response · Author response to Decision Letter 0]

4 Mar 2022

We have responded to specific reviewer and editor comments in our 'Response to Reviewers' file.

---

## [Editor Report · Decision Letter 1]

14 Mar 2022

Prediction of malignant transformation in oral epithelial dysplasia using infrared absorbance spectra

PONE-D-21-39833R1

Dear Dr. Shaw,

We’re pleased to inform you that your manuscript has been judged scientifically suitable for publication and will be formally accepted for publication once it meets all outstanding technical requirements.

Kind regards,

Patrick Ha

Academic Editor

PLOS ONE

Additional Editor Comments (optional):

The authors have done a good job in responding to the reviewers' comments. This is a complex topic, and the algorithm(s) outlined are at the very least intriguing and warrant further study.
---

## [Editor Report · Acceptance letter]

18 Mar 2022

PONE-D-21-39833R1 

Prediction of malignant transformation in oral epithelial dysplasia using infrared absorbance spectra 

Dear Dr. Shaw:

I'm pleased to inform you that your manuscript has been deemed suitable for publication in PLOS ONE. Congratulations! Your manuscript is now with our production department. 

Kind regards, 

on behalf of

Dr. Patrick Ha 

Academic Editor

PLOS ONE